# The Influence of the Structure of Cotton Fabrics on the Adhesion of Conductive Polymer Printed with 3D Printing Technology

**DOI:** 10.3390/polym15030668

**Published:** 2023-01-28

**Authors:** Rocio Silvestre, Eduardo Garcia-Breijo, Josué Ferri, Ignacio Montava, Eva Bou-Belda

**Affiliations:** 1Instituto Tecnológico del Textil (AITEX), 03801 Alcoy, Spain; 2Instituto Interuniversitario de Investigación de Reconocimiento Molecular y Desarrollo Tecnológico (IDM), Universitat Politècnica de València, 46022 Valencia, Spain; 3Instituto Educación Secundaria Poeta Paco Mollà, Av. Reina Sofía s/n, 03610 Petrer, Spain; 4Department of Textile and Paper Engineering, Universitat Politècnica de València, Plaza Ferrándiz y Carbonell s/n, 03801Alcoy, Spain

**Keywords:** functional textile, 3D printing, adhesion, smart textile, conductive, heater

## Abstract

Three-dimensional printing technology is being increasingly applied in a multitude of sectors. However, this technology is not generally applied in the same way as in other sectors, possibly due to the difficulty of adhesion between the polymer and the textile substrate. A textile garment is subjected to wear and tear during its lifetime, and a low tensile strength or rubbing resistance hinders a garment in most of the applications of this type of research. This study examined the influence of the characteristics of the cotton textile substrate, such as the weave structure and the yarn thickness, on the tensile strength of a 3D-printed element with conductive filament. Starting from the fabric with the highest tensile strength, different prints were made using this technology to incorporate conductive and heating properties into the fabric. The results validate the possibility of providing new properties to the textile by means of this technology; however, the correct selection of the textile used as a base substrate is important.

## 1. Introduction

Recently, there have been great advances in 3D printing, so called because, in addition to printing on the xy plane, as with other printing technologies, it is also capable of superimposing prints on the z plane. This is why this technology is also known as additive manufacturing (AM), because it allows the progressive adding of layers of material [1,2]. Possibilities for customization, rapid prototyping, waste reduction, and design freedom to create complex structures [3] are some of the main benefits of AM or 3D printing [4,5]. Many 3D printing techniques use low-melting-temperature polymeric materials [6,7,8], but it is not only the plastics industry that can take advantage of this technology; other complex areas and sectors can also benefit, such as construction [9,10,11,12], the automotive industry [13,14,15], the metal industry [16], regenerative medicine [17,18,19,20], and musical instrument fabrication [21,22], among others [23,24,25]. The textile industry has not remained on the sidelines either, since the advantages of customization, the use of complex designs, and the wide range of materials that can be printed make new products and innovations possible [26].

Analyzing the different applications, we can see that, in addition to printing techniques, some of the advances in this type of printing are also largely thanks to the possibilities of using almost any type of design [27]. In the textile industry, much work has been carried out on the creation of articles or garments that could be manufactured with conventional 3D printers. There have been a range of studies presenting different designs that allow the industrial process to be accelerated to manufacture garments by printing small modular pieces with materials such as ABS, PLA, or TPU. These pieces have adequate mechanisms so that they can be assembled with each other and that also offer some flexibility to facilitate their integration into the garment [28]. Other authors investigated the effects produced by 3D printing on textiles in order to customize or personalize garments. Aspects such as the levels of adhesion and durability in washing have been studied [8], or even complex printing with various geometries as if they were fabrics, evaluating flexibility and stretchability [29]. At the same time, other articles explore the possibility of introducing 3D-printed parts in the commercial production of garments, evaluating the costs and time required. They also analyze the weak points in each of the parts of the manufacturing process which limit mass production [30].

On the other hand, some of the advances made in this type of printing are also thanks to the possibilities of using almost any type of material. The use and selection of the materials can also have a great impact on the industry. There is research on an increasing number of materials, but the most relevant ones with direct application in the textile world are the following. Materials such as PLA (polylactic acid) have been developed, which is one of the most important environment-friendly biodegradable thermoplastic polyesters with considerable applicability in textiles from the point of view of sustainability [31,32]. In addition, there are an infinite number of materials, such as nanomaterials and materials with additives or compounds, with properties that can add great value to the product. In the electronics world and wearables, there are several issues to be resolved so that the integration of the electronics world with garments offers a complete solution. Although electronics are becoming smaller and their integration more flexible, it would be ideal to be able to embed these electronics in the textile as if it were a single element. In this sense, there is another very special group of materials with electrical and conductive properties that can facilitate the integration of sensors or small electronic components that contribute to the development of smart textiles [33,34,35].

The objective of this study was to examine the adhesion of PVA material printed by FDM 3D printing technology on cotton/polyester fabrics with different structural characteristics. The influence of the adhesion strength of the printed polymer on three fabrics with different bonds, varying the bonding coefficient, and using two yarns of different thicknesses of cotton inserted in the weft direction was evaluated. With the fabric that showed the best adhesion, conductivity and heating measurements were carried out in order to validate the 3D printing process to obtain conductive textiles.

## 2. Materials and Methods

For this research, the fabric samples used are listed in Table 1. Warp density was 60 threads/cm, and the warp yarn was a tangled multifilament PES 167 dtex/48 filaments. Two different sets of weft yarns with different thicknesses were used, 50 Nm (fine thread) and 15 Nm (thick thread), both single-ply cotton yarns woven using open-end technology (supplied by R. Belda Llorens S.A, Banyeres de Mariola, Spain). Each weave pattern was designed to reach maximum weft density. The samples were produced using a Smit GS 900 weaving machine of 190 cm width, with a Staubli DX-100 electronic Jacquard machine and EAT DesignScope software.

Figure 1 depicts the samples’ rapport, and Figure 2 displays a three-dimensional representation of these fabrics. Figure 1a and Figure 2a show the taffeta, the smallest course weave pattern possible, where it can be observed that the number of interlacing points amounts to 4. In the images corresponding to 2 e 1 twill and 3 e 2 satin, the course of the weave pattern increases, and so does the number of interlacing points; however, the interlacing coefficient (IC) decreases.

Currently there are no standardized tests to determine the tensile adhesion forces between a textile substrate and a polymer, so in order to know this value a customized test protocol has been established. This measurement protocol, described by the authors of [36], is based on the design of two 3D pieces that allow the polymer and the fabric to be held in the dynamometer. The part that was clamped in the upper jaw was printed, using FDM printing technology, directly on the fabric under study. While the part that was placed in the lower jaw was designed and 3D-printed with the aim of housing and holding the fabric on a stable base.

The upper piece, which was printed directly on the fabric and consisted of a circular base of 12 mm radius, will be the contact surface with the fabric; therefore, it will be the only area where the fabric is adhered to the conductive PLA. Moreover, the dimensions of the gripping area of the piece with the clamp are critical; both the height and the thickness are key; it cannot be too low or too thin because the closing pressure of the clamp will split the piece. It should also be noted that the higher the part, the higher the probability of printing failure, so several versions of the part were designed until the most suitable dimensions were found.

Both parts were designed using the SolidWorks computer-aided design (CAD) program. As can be seen in Figure 3, the -D model has an outer line in the form of a square that optimizes the alignment of the upper part with the lower one.

The part to be placed in the lower jaw was designed consisting of a base for the fabric and a gripping area for the jaw and the fabric. On this piece will be placed the piece described in Figure 4, together with the analyzed textile. Specifically, the design was made following the measurements of the dynamometer jaw and consisted of a T-shaped piece. This piece allowed the fabric to be held in the lower jaw. Moreover, the dimensions of the base coincided with the square printed on the fabric in order to center both 3D pieces in the vertical axis of the dynamometer. Specifically, the part had the shape shown in the attached image extracted from the design made using SolidWorks. Variations were also made of this part until the optimal design for this application was achieved.

The 3D printing process on the textile substrate consisted of printing the part described in the previous section directly on the fabric. The printer used was an Anet ET4, a fused deposition modeling (FDM) 3D printer that consisted of a metal chassis with a 32-bit base plate. The printer features were as follows:−Maximum print size: 220 × 220 × 250 mm^3^.−Print resolution: ±0.1 mm.−Maximum extruder temperature: 250 °C.−Maximum temperature of the printing bed: 100 °C.−Filament diameter: 1.75 mm.

Table 2 summarizes the printing characteristics used.

The parameters of the 3D printing system used, which are shown in Table 2, were determined by following the optimal conditions used as a result of the work reported by Spahiu et al. [29]. To print on the textile, the first step was to place the fabric on the printing bed. To achieve the best possible results and the greatest possible stability in the results, the positioning of the fabric was decisive. To do this, lacquer was used to prevent the fabric from slipping during the printing process, and then it was stretched and fixed on the base using tweezers.

The height of the printing bed was then leveled. The Anet ET4 printer has a pressure sensor that is placed in the nozzle and allows the bed to be automatically leveled. This process was carried out every time the type of fabric was changed. Finally, the file configured in the Cura Ultimaker slicer was selected for the printer to start the printing process.

The dynamometer used during the polymer–substrate bond strength test was a Zwick/Roell Z005 desktop dynamometer controlled by Zwick’s TestXpert V10.11 software (Zwick, Ulm, Germany).

Once the alignment of the grips was set, the initial height (LE) between the grips at the start of the test was established. This height was determined according to the thickness of the support base of the lower jaw part and the dynamometer grip area of the printed part on the textile. The initial separation distance between the two load cells was 21.948 mm.

Moreover, a preload of 2N was considered so that all the tests started taking data from a certain point.

Once the starting conditions for the test had been determined, the test was performed, the procedure for all tests was as following: Firstly, the textile substrate with the 3D part was placed on the base part, aligning the outer square printed on the substrate with the base of the lower jaw part. Subsequently, the sides of the T-shaped part were covered with the excess fabric and the lower jaw was closed.

The upper jaw was then lowered to the LE position and the upper jaw was closed.

Table 3 summarizes the tensile test conditions used in the dynamometer.

Samples were characterized using an optical microscopic, and images were taken with a LEICA MZ APO stereomicroscope, which was used to analyze the penetration of the polymer into the tissue by analyzing the cross section of the fabric.

Once the fabric with the best adhesion results was determined, the electrical resistivity of the conductive PLA with graphene (BlackMagic3D) was measured. It was determined both raw with the filament taken directly from the spool and annealed in an FED-115 oven (Binde GmbH, Millvilem, NJ, USA) at 220 °C for 10 s. The resistivity measurement was performed using a 4200A-SCS Semiconductor Characterization System (Keythley, Solon, OH, USA) from which the resistance value of 3 filament specimens of 20, 40, and 80 mm with a diameter of 1.78 mm was obtained (Figure 5). The resistivity was obtained from the relationship between the resistivity and resistance, length, and area of the filament.

The resistivity of a disk fabricated by the 3D printer was also measured using the Van der Pauw or 4-prong technique [37,38,39]. For this measurement, a 72-13300 DC power supply (TENMA, Tokyo, Japan) was used, measuring simultaneously the voltage and current with 2 SDM3045X digital multimeters (SIGLENT, Augsburg, Germany). The measurement was performed on a 40 mm and 0.7 mm thick disk (Figure 6) which had 4 zones to apply the 4-point method.

Finally, a heater in coil format was printed on a COR22 fabric with the dimensions shown in Figure 7. A thermal study of the heater in operation was carried out using an 875 thermal camera (Testo SE & Co. KGaA, Lenzkirch, Germany).

## 3. Results

Table 4 shows the results of the tensile adhesion force obtained by dynamometry between the printed device and each of the fabrics used, using three different structures (three simple ligaments of different courses and binding coefficients) and varying the count or thickness of the cotton yarn used in the weft direction. Five tests were performed for each fabric sample and their average was calculated.

These same results detailed in Table 4 are represented in Figure 8, where the adhesion obtained is compared according to the weave of the fabric used and the thickness of the yarn. A clear difference is observed between fabrics whose wefts are of different thicknesses, with the same behavior regardless of the weave present in the fabric. The higher the yarn count (or thickness), the greater the adhesion of the printed conductive polymer on the textile substrate. Similarly, there is a significant influence between the weave used and the adhesion achieved. A better adhesion in the satin fabric and a lower adhesion in the taffeta fabric were achieved.

Table 5 shows the binding coefficient of each of the samples and the force obtained. These same results are represented in Figure 7, which shows the influence of the bonding coefficient of each of the textiles tested on the resulting adhesion force.

Figure 9 shows a clear relationship between the bonding coefficient and the adhesion obtained between the polymer and the surface of the fabric, showing that the higher the bonding coefficient, the lower the adhesion. This decreasing tendency of adhesion is observed in both cases, when using thick or thin yarn, but as it is also observed in both Figure 8 and Figure 9, the adhesion of both materials is greater when using thicker yarn as weft. There is a difference of between 30 and 40 N if the results are compared between the thick and thin yarn of the three fabrics with different weaves.

To understand the difference between the weaves used, a simulation of the three fabric structures studied is shown in Figure 2, a 3D representation of the surface of the fabric, and in Figure 10, a representation of each of these structures in cross section, where the interweaving of the warp with the weft and the evolution of a thread from the face of the fabric to the back can be clearly observed.

The 3D representation of each ligament (Figure 2) shows the evolution of the warp yarns with respect to the weft yarns, which are cross-sectioned. In these, the following can be observed:−In the taffeta weave, each warp yarn makes an inflection point from the face to the underside of the fabric in every other pass.−In the case of the twill weave, the inflection of each thread to evolve from the face to the underside of the fabric occurs every three passes.−In the case of the satin weave, the inflection occurs every five passes.

These evolutions occur with both fine and coarse wefts, with the difference that the inter-row spaces between the coarse wefts are much larger than those of the fabric made with fine wefts. These inter-weft spaces offer a greater contact surface between the polymer and the textile substrate, taking into account that the yarn used in the warp direction was the same in all the fabrics developed.

Figure 11 shows the magnified images of the cross section of the Cos32 and Cos20 fabrics printed using 3D technology. In Figure 11a, where a thicker weft yarn is used, the interaction between yarns and the printed polymer can be clearly observed. Focusing on the depth of penetration of the polymer in the interfilament gaps, it is observed that at those points of yarn inflection, the penetration of the conductive polymer is lower, between 0.090 and 0.180 nm. On the other hand, in the interfilament spaces where there is no warp inflection point and two weft yarns are found together, there is a greater depth of penetration of the printed polymer, with a depth of 0.270 nm. If we compare the penetration behavior of the conductive PLA in the fabric with a thread of lesser thickness in the weft direction (Figure 11b), the printed polymer is not in contact with the surface of the thread, with a space between the polymer and the surface of the fabric. It should be noted that, in the same way as in the fabric of greater yarn thickness, at the points where the inflection point is found, there is less depth, with a penetration of 0.02 nm, while in the spaces where the warp yarn inflection is not found, there is a depth of 0.04 nm. It is worth highlighting the difference found in the depth of penetration of the polymer in the enclaves formed between the weft yarns, which was 0.270 nm in the weft fabrics of thick yarn and 0.04 nm in the weft fabrics of finer yarn.

Once the fabric with the best adhesion between the polymer and textile substrate was known, which is the COR22 fabric, an impression was made to determine the resulting conductivity.

Table 6 and Figure 12 and Figure 13 show the results of the resistivity measured on 20, 40, and 80 mm standard filaments in both raw and annealed conditions. It is observed that the measurements on the 20 mm specimens are slightly different from the 40 and 80 mm specimens. The resistivity before extrusion is 0.0136 Ω-m, decreasing after extrusion to 0.0105 Ω-m for the 80 mm filaments. The manufacturer supplies a resistivity of 0.0060 Ω-m, without specifying whether this value corresponds to its raw or already extruded state.

With the data obtained using Van der Pauw’s technique for an already extruded part, a resistivity of 0.0222 ± 0.0022 Ω∙m was established.

The designed heater has a track width-to-length ratio of L/W = 146.64, so the theoretical resistance, with the resistivity measured by Van der Pauw, is determined by Equation (1):(1)$$R=\rho \frac{L/W}{t}=0.0222\cdot \frac{146.64}{0.001}=3255.48\;\Omega$$

The measurement made by the Kelvin method at four wires gives a result of 3201 Ω, so it can be stated that the material has an approximate resistivity of 0.0222 Ω∙m.

To confirm its use as a heater, a constant voltage was applied up to about 40 °C, which happened after approximately half a minute (Figure 14).

## 4. Discussion

The results show that the correct selection of the characteristics of the textile substrate used as a base in 3D printing would increase the application possibilities of this technology to obtain innovative high-performance textiles. The number of yarns and the structure of the fabric have an influence. By visualizing the cross section of the printed samples, it was possible to observe the penetration of the polymer between the yarn interstices. With a higher yarn thickness, the gaps between yarns are larger and a higher penetration of the printed polymer can be obtained and thus a larger contact surface between the substrate and filament, which leads to higher adhesion. This same gap between the yarns arranged side by side is reduced when the warp yarn passes through the middle of both yarns, passing from the right side to the reverse side or vice versa (inflection point), so that the more inflection points there are in the fabric, the lower the adhesion between the textile and the printed filament.

If we focus attention on the results in Table 4, which shows the adhesion strength between the printed piece and the different fabrics tested, it can be seen that the satin structure fabric obtains greater adhesion in both cases, regardless of whether a thicker or thinner yarn is used. This is due to the inflection points commented on previously and is related to the fabric’s binding coefficient, which is lower (IC = 0.4). The adhesion of the print on the substrate decreases when the structure of the fabric used has a higher binding coefficient, going from 70.49 N with a satin weave (IC = 0.4) to 35.69 N with a twill weave (IC = 0.6) and even lower when the weave is taffeta (IC = 1). The same behavior is observed when a thicker yarn is used, although greater adhesion is achieved in the three structures.

Once the fabric with the highest adhesion was obtained, different conductivity and heating tests were carried out. First, resistivity measurements were made on standard filaments of 20, 40, and 80 mm, and it can be seen in Table 6 that the resistivity is slightly higher in the measurements on the narrower 20 mm specimens. This may be possible due to filament imperfections that become more evident when the sample size is smaller.

If the resistivity results before and after extrusion are compared, in all cases the resistivity is higher, the polymer losing conductivity with printing. To confirm its use as a heater, the material was subjected to a constant voltage. Figure 14 shows how the test begins with the polymer showing a temperature of 26 °C which reaches 39 °C after 25 s.

## 5. Conclusions

The results show a significant difference in the adhesion between fabric and polymer, depending on the fabric structure and yarn thickness. If a fabric in which a weft yarn of lesser thickness is used, there is less adhesion of the conductive polymer. Moreover, the adhesion also varies according to the bonding coefficient of the fabric weave structure, with an increase in adhesion with a lower bonding coefficient. The use of this printing technology to obtain smart textiles is validated in this work, and good results in terms of conductivity and heating properties can be obtained.

## Figures and Tables

**Figure 1 polymers-15-00668-f001:**
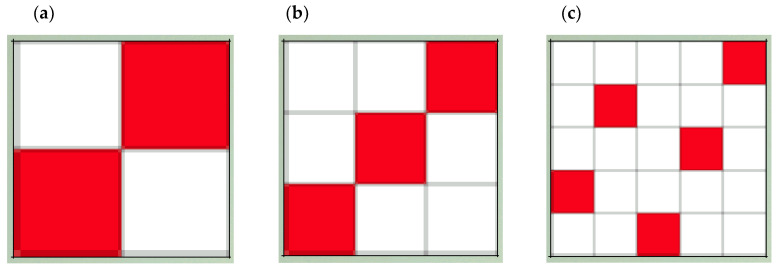
Weave diagrams of fabrics used: (**a**) taffeta 1 e 1; (**b**) twill 2 e 1; (**c**) satin 3 e 2.

**Figure 2 polymers-15-00668-f002:**
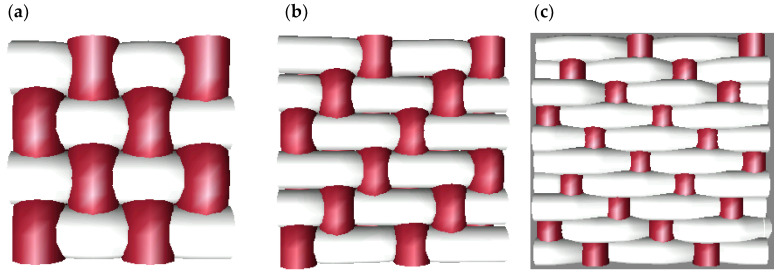
Three-dimensional simulation of fabrics used: (**a**) taffeta 1 e 1; (**b**) twill 2 e 1; (**c**) satin 3 e 2.

**Figure 3 polymers-15-00668-f003:**
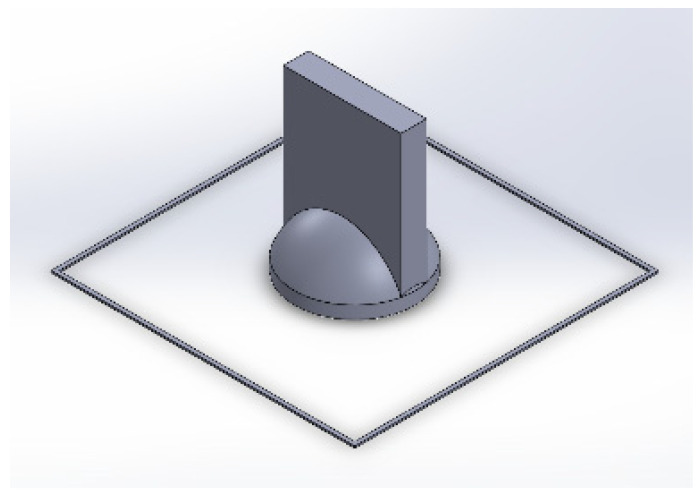
Three-dimensional CAD model using SolidWorks.

**Figure 4 polymers-15-00668-f004:**
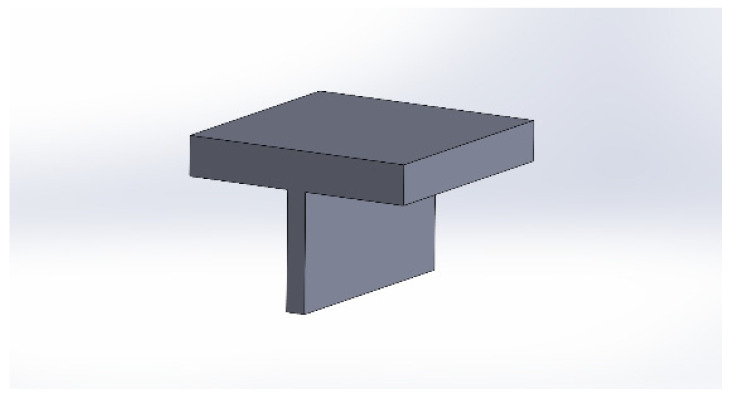
Part for dynamometer in SolidWorks.

**Figure 5 polymers-15-00668-f005:**
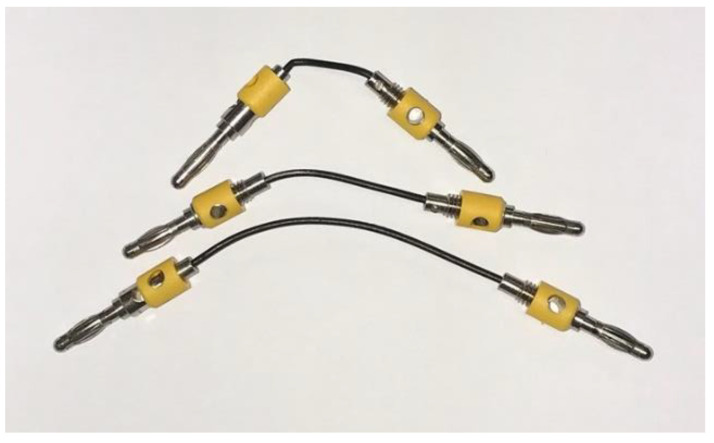
The 20, 40, and 80 mm test specimens connected to the banana probes.

**Figure 6 polymers-15-00668-f006:**
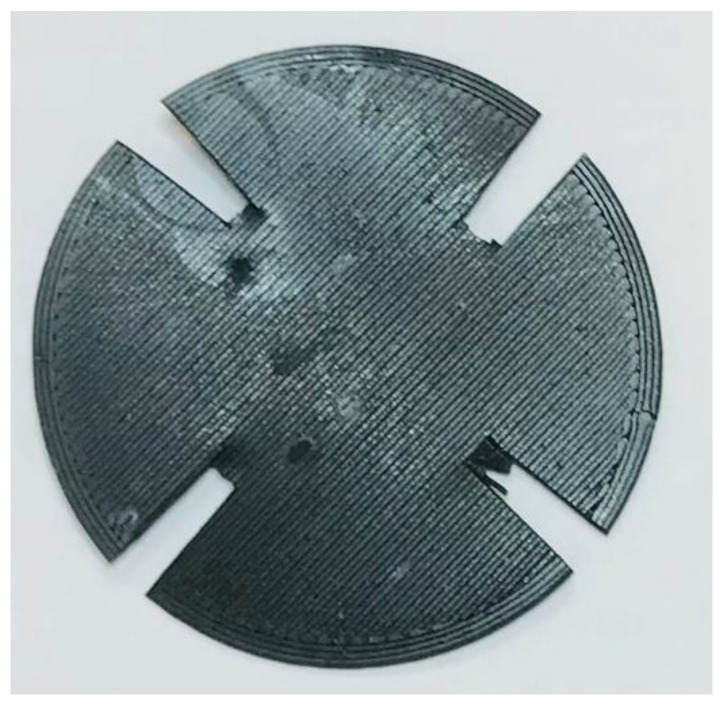
The 40 mm disk prepared for the Van der Pauw technique.

**Figure 7 polymers-15-00668-f007:**
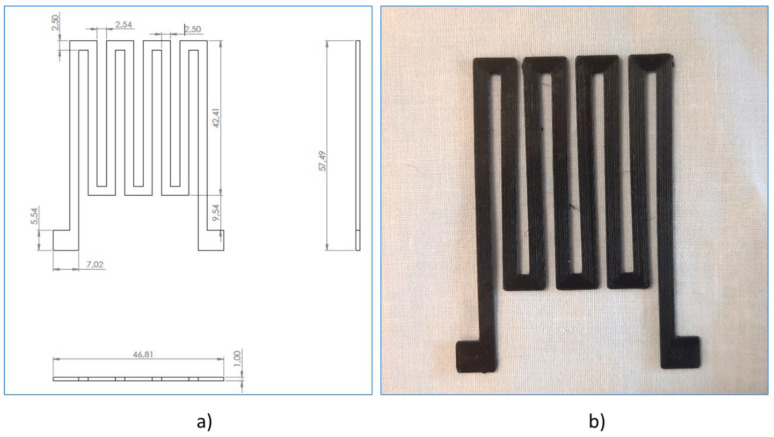
(**a**) Heater measurements; (**b**) heater printed on COR22 fabric.

**Figure 8 polymers-15-00668-f008:**
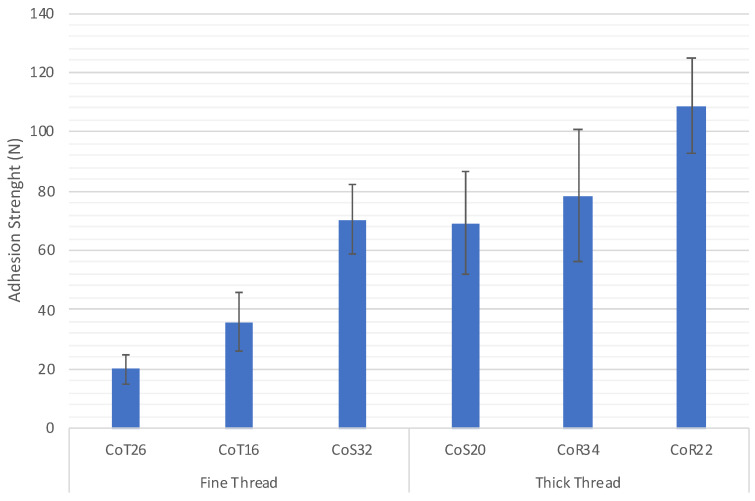
Adhesion strength between fabric and polymer printed.

**Figure 9 polymers-15-00668-f009:**
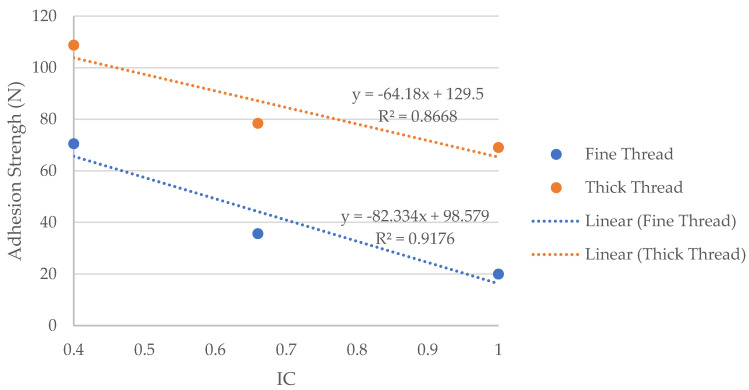
Graph representation of the relationship between adhesion strength and binding coefficient of the printed fabrics.

**Figure 10 polymers-15-00668-f010:**
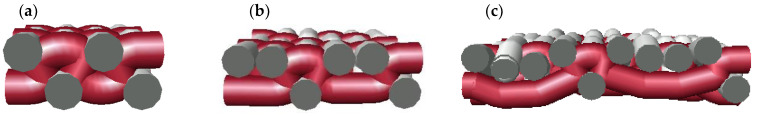
Cross-sectional 3D simulation view of fabrics used: (**a**) taffeta 1 e 1; (**b**) twill 2 e 1; (**c**) satin 3 e 2.

**Figure 11 polymers-15-00668-f011:**
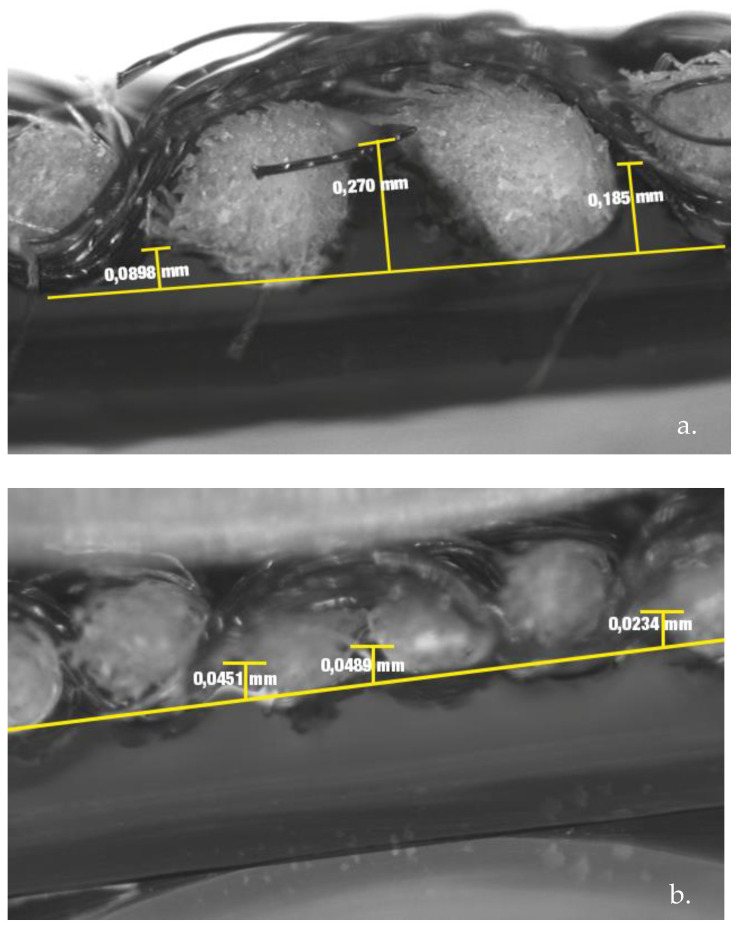
Magnified images (40x) of cross sections of the fabrics: (**a**) Cos20 (thick thread) and (**b**) Cos32 (fine thread).

**Figure 12 polymers-15-00668-f012:**
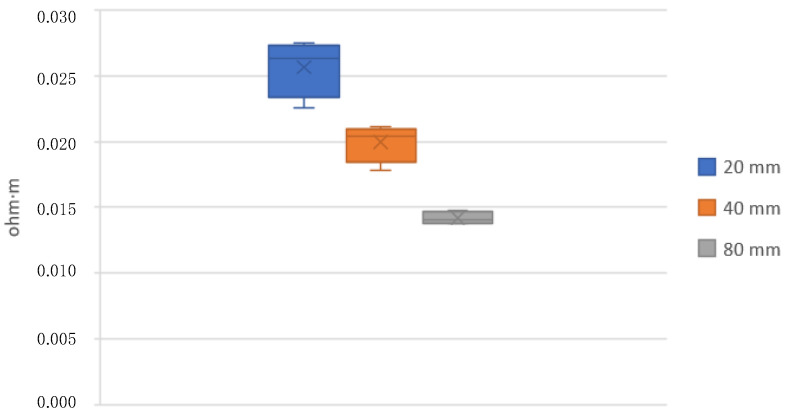
Raw filament resistivity.

**Figure 13 polymers-15-00668-f013:**
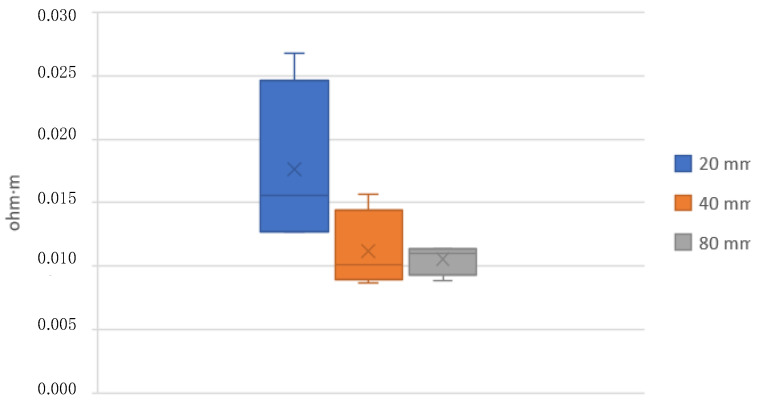
Annealed filament resistivity.

**Figure 14 polymers-15-00668-f014:**
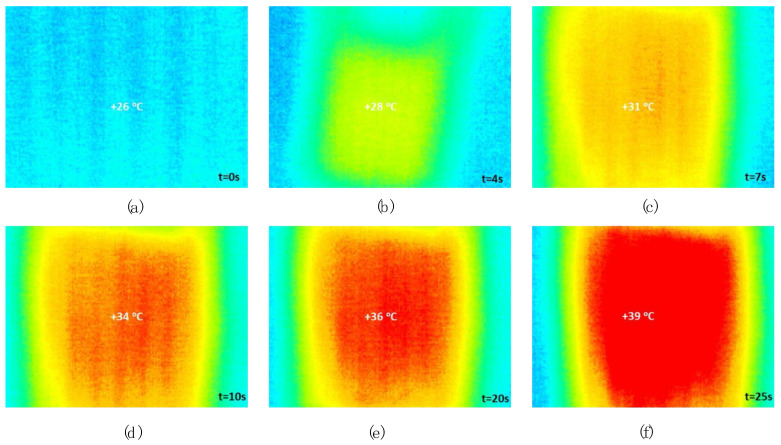
Time evolution of the temperature of the heater printed on the fabric. (**a**) 0 sec, (**b**) 4 sec, (**c**) 7 sec, (**d**) 10 sec, (**e**) 20 sec, (**f**) 25 sec.

**Table 1 polymers-15-00668-t001:** Characteristics of fabric samples developed.

Reference	Thread Thickness (Nm)	Weave	Course	Rapport	IC	Weft Density (threads/cm)
CoT26	50.00	Taffeta	1 × 1	1 e 1	1	26
CoS32	50.00	Twill	3 × 3	2 e 1	0.66	32
CoR34	50.00	Satin	5 × 5	3 e 2	0.4	34
CoT16	15	Taffeta	1 × 1	1 e 1	1	16
CoS20	15	Twill	3 × 3	2 e 1	0.66	20
CoR22	15	Satin	5 × 5	3 e 2	0.4	22

**Table 2 polymers-15-00668-t002:** Printing parameters used.

Printing Parameters	
Conductive material	PLA graphene (Black Magic 3D)
Printing temperature	220 °C
Bed temperature	100 °C
Layer height	0.2 mm
Initial layer height	0.2 mm
Printing speed	50 mm/s
Printing speed initial layer	25 mm/s

**Table 3 polymers-15-00668-t003:** Tensile test parameters.

Basic Test Parameters	
Preload	2 N
LE distance	21,948 mm
Test speed	5 mm/min

**Table 4 polymers-15-00668-t004:** Results of adhesion strength obtained by traction between the printed device and the substrate.

Reference Sample	Strength (N)	Variance
**CoT26**	19.5	76.59
**CoS32**	35.69	101.38
**CoR34**	70.49	62.64
**CoT16**	69.1	60.96
**CoS20**	78.43	22.18
**CoR22**	108.77	14.79

**Table 5 polymers-15-00668-t005:** Adhesion strength and bonding coefficient of the printed fabrics.

Reference Sample	Strength (N)	IC
**CoT26**	19.95	1
**CoS32**	35.69	0.66
**CoR34**	70.49	0.4
**CoT16**	69.1	1
**CoS20**	78.43	0.66
**CoR22**	108.77	0.4

**Table 6 polymers-15-00668-t006:** Results of resistivity measured.

Sample	Length (mm)	Raw Filament	Annealed Filament
		Resistivity (Ω·m)	Resistivity (Ω·m)
**1**	20	0.0225	0.0126
**2**	20	0.0258	0.0127
**3**	20	0.0269	0.0184
**4**	20	0.0275	0.0267
**5**	40	0.0179	0.0087
**6**	40	0.0204	0.0095
**7**	40	0.0212	0.0107
**8**	40	0.0204	0.0156
**9**	80	0.0148	0.0112
**10**	80	0.0138	0.0113
**11**	80	0.0138	0.0088
**12**	80	0.0144	0.0106

## Data Availability

Not applicable.

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
