# Peer review of "The Influence of the Structure of Cotton Fabrics on the Adhesion of Conductive Polymer Printed with 3D Printing Technology"

_polymers, 2023, doi:10.3390/polym15030668_

Round 1

Reviewer 1 Report

The manuscript reported by Silvestre et al entitled as “Study of the influence of the structure of cotton fabrics on the adhesion of conductive polymer printed with 3D printing technology” attempts to study the influence of the cotton textile substrate, such as the weave structure and the yarn thickness on the adhesion of the 3D printed conductive PLA. This study is important as it explores parameters which can improve the adhesion between 3D printed polymers and textile scaffolds. However, the manuscript still needs some more revision before the final acceptance. My specific comments about the manuscript are follows.

1.      The present study only considers the effect of the weave structure and the yarn thickness on the adhesion of 3D printed PLA. However, many other important 3D printing parameters such as z-distance (the distance between the nozzle and the printing bed), the temperature of the nozzle of the printer could be very important for influencing the adhesion on the textile scaffolds. Hence, in my opinion, authors should study these parameters also to make the paper more robust and informative.

2.      Figure 7 does not show any standard deviation, so please include the standard deviation and also mention the total number of samples used for testing.

3.      The discussion section of the manuscript seems to be too short; authors should consider expanding this section by providing more detailed interpretations of the results obtained.

4.      Finally, as a minor comment, please include a section called statistical analysis in the materials and methods section and mention clearly the statistical method used to validate the significance of the results obtained with the adhesion measurements.

………………………………………………………………………………………

Reviewer 2 Report

Please see attached pdf.

Round 2

Reviewer 1 Report

Authors have satisfactorily addressed all of my concerns, so the manuscript can now be accepted in the present format 

Reviewer 2 Report

Authors have made a great job concerning the revisions asked. I happily agree with the manuscript's acceptance for publication and wish the best to the authors.